# Optimal iron content in ready-to-use therapeutic foods for the treatment of severe acute malnutrition in the community settings: a protocol for the systematic review and meta-analysis

Aamer Imdad ,[1] Melissa François,[2] Fanny F Chen,[2] Abigail Smith,[3] Olivia Tsistinas,[3] Emily Tanner-Smith ,[4] Jai K Das ,[5] Zulfiqar Ahmed Bhutta[6]

For numbered affiliations see end of article.

**Correspondence to**
Dr Aamer Imdad;
imdada@upstate.edu

## ABSTRACT

**Introduction** The current standard of care for children with severe acute malnutrition (SAM) involves using ready-to-use therapeutic food (RUTF) to promote growth; however, the precise formulation to achieve optimal recovery remains unclear. Emerging research suggests that alternative RUTF formulations may be more effective in correcting SAM-related complications such as anaemia and iron deficiency. This systematic review commissioned by the WHO aims to synthesise the most recent research on the iron content in RUTF and related products in the community-based treatment of uncomplicated severe malnutrition in children aged 6 months and older.

**Methods and analysis** We will search multiple electronic databases. We will include randomised controlled trials and non-randomised studies with a control arm. The intervention group will be infants who received RUTF treatments other than the current recommended guidelines set forth by the WHO. The comparison group is children receiving RUTF containing iron at the current WHO-recommended level of 1.9 mg/100 kcal (10–14 mg/100 g). The primary outcomes of interest include blood haemoglobin concentration, any anaemia, severe anaemia, iron-deficiency anaemia, recovery from SAM and any adverse outcomes. We will use meta-analysis to pool findings if sufficient homogeneity exists among included studies. The risk of bias in studies will be evaluated using the Cochrane risk of bias-2. We will use the Grading of Recommendations Assessment, Development, and Evaluation(GRADE) approach to examine the overall certainty of evidence.

**Ethics and dissemination** This is a systematic review and will not involve direct contact with human subjects. The findings of this review will be published in a peer-reviewed journal and will guide the WHO's recommendation on the optimal iron content in RUTFs for the treatment of SAM in children aged 6–59 months.

## BACKGROUND

The WHO estimates that over 45 million children under the age of five worldwide suffered from wasting (low weight for height) in 2020

### Strengths and limitations of this study

► This systematic review commissioned by the WHO will synthesise the most recent research on iron content in ready-to-use therapeutic foods and related products in the community-based treatment of uncomplicated severe malnutrition in children aged 6 months and older.

► We will search several databases for relevant literature and include randomised and non-randomised studies.

► We will assess risk-of-bias for each outcome and use the Grading of Recommendations Assessment, Development and Evaluation to assess the overall quality of evidence.

► We will assess the following outcomes: blood haemoglobin concentration, any anaemia, severe anaemia, iron-deficiency anaemia, recovery rates, all-cause mortality, adverse events, growth outcome and serum levels of micronutrients.

► We will conduct meta-analyses if data are available from more than one study and there is clinical and methodological homogeneity in the included studies.

alone.[1] Children with severe acute malnutrition (SAM) have about sixfold increased mortality risk compared with well-nourished children.[2] In addition to the risk of mortality, children with SAM are at increased risk of morbidities such as diarrhoea, pneumonia, measles and micronutrient deficiencies, and long-term neurodevelopmental delay.[2–4] Nutritional rehabilitation of children with SAM is a key intervention that helps treat morbidity and prevent mortality.[3 5 6] The current standard of care for children with uncomplicated SAM involves using ready-to-use therapeutic food (RUTF) to promote growth[7]; however, the precise formulation to achieve optimal recovery remains unclear.[6 8] There are ongoing efforts to improve RUTF

to optimise the composition and balance of macronutrients and micronutrients.[9–11]

SAM involves deficits of several key macronutrients and micronutrients, with studies showing higher rates of anaemia in children suffering from malnutrition.[12–14] The prevalence of anaemia in severely acutely malnourished children ranges from 40% to 90%, and about half of the anaemia is attributed to iron deficiency.[15] Iron deficiency in severely malnourished children could be due to low intake, increased losses, higher demand and poor absorption. Iron is essential for adequate catch-up growth and neurological development in children with SAM.[12 16 17] Moreover, recent studies suggest alternative RUTF formulations may be more effective in correcting anaemia and iron deficiency in children with SAM compared with current standard RUTF formulations.[12 18 19] In addition to the need for sufficient iron intake to treat iron deficiency, it remains unclear whether improving the iron status among the undernourished children increases the risk of complications, including the risk of morbidities such as diarrhoea, malaria and undesired changes in the microbiome.[20–22] This aspect is especially of concern in malaria-endemic regions in Africa because nearly a third (27%) of all children affected by wasting worldwide reside in Africa.[22] Due to the critical role of RUTF in the medical management of children with SAM, more information is needed to generate formulations with optimal iron levels for treatment of anaemia and optimal growth and development in children suffering from malnutrition.[16–18] A number of recent studies have been completed to assess the optimal dose of iron in RUTF in the community management of SAM, and there is no existing systematic review on this topic.[12 19 23] Furthermore, WHO has started to synthesise the evidence to update the current guidelines for treating wasting in children. Therefore, this WHO commissioned review aims to synthesise the most recent research on the iron content in RUTF and related products in treating severe malnutrition in children 6–59 months of age.

## OBJECTIVE
### Primary objective
In children aged 6 months or older with uncomplicated SAM being treated in the community settings, does increased iron dose in RUTF compared with the WHO recommendation for iron fortification of RUTF improve outcomes such as blood haemoglobin concentration, recovery from iron-deficiency anaemia, and so on?

### Secondary objective
In children aged 6 months or older with uncomplicated SAM being treated in the community settings, what mechanisms other than increased dose of iron in RUTF (such as type, composition, zinc, phytate content, etc.) can help increase the bioavailability of iron and improve the recovery from iron-deficiency anaemia?

## METHODS AND ANALYSIS
We will follow the standard guidelines of the Cochrane Collaboration to conduct the systematic review. We registered a detailed protocol on PROSPERO (ID=CRD42021278006) and would report the systematic review findings according to Preferred Reporting Items for Systematic Reviews and Meta-Analyses 2020 guidelines.[24]

### Study type
We will consider randomised trials, and that may include individual and cluster randomised trials. We will also consider non-randomised studies if there are not enough randomised studies. The data will be analysed separately for randomised and non-randomised studies. We will include the following non-randomised studies: non-randomised controlled clinical trials and controlled before and after (CBA) studies. Case–control studies, interrupted time series, programme evaluations, case reports, case series and commentaries will be excluded. The definitions of eligible studies design and consideration for inclusion in this review are available in table 1.

### Population
The population of interest is children aged 6 months or older with SAM managed in outpatient settings. The definition of SAM will be based on weight for height Z scores (< −3 SD for WHO growth standards) or mid-upper arm circumference (<115 mm) or presence of bilateral oedema.[25] We will consider studies that included children who were previously admitted to the hospital and are now being rehabilitated in community settings with the help of RUTF. We will exclude studies of children with complicated SAM who are admitted to the hospital, as those participants might have complications such as pneumonia or severe diarrhoea. We will exclude studies done specifically on participants who have chronic diseases, genetic disorders, or congenital anomalies.

### Intervention
The intervention of interest is the dose of iron in RUTF other than in the standard RUTF for the treatment of SAM in community settings. We will include studies of iron-fortified RUTF and exclude those studies where iron was given separately as supplements, such as tablets, syrup, multiple micronutrient powder, and so on. We will include studies where RUTF is given as the main intervention, and it met total daily requirements or was given as a supplementary intervention to the usual diet for children with SAM recovering at home. WHO recommends that a standard RUTF should have at least 1.9 mg/100 kcal (or 10–14 mg/100 g) of iron to treat SAM in children 6–59 months of age.[25] We will include studies irrespective of the type of RUTF used, that is, standard RUTF versus low milk-based versus non-milk-based versus locally prepared. We will exclude studies where RUTF was given to healthy children to prevent SAM; for treatment of moderate acute malnutrition; and as part of complementary feeding.

**Table 1** The definitions of eligible study designs

| Study designs | Definition | Notes |
|---|---|---|
| RCT or randomised trial | 'An experimental study in which people are allocated to different interventions using methods that are random'. | We will include both individual and cluster randomised trials. In individual randomised trials, the randomisation is done at the individual levels, while in cluster randomised trials, the randomisation is based on cluster or groups of individuals. We will also consider factorial design trials where multiple interventions are studied in the same trial. |
| NRCT or non-randomised trial | 'An experimental study in which people are allocated to different interventions using methods that are not random'. | We will avoid using the term quasiexperimental studies as it means differently by different authors. We will exclude the experimental studies where there was no control group. |
| CBA study | 'A study in which observations are made before and after the implementation of an intervention, both in a group that receives the intervention and in a control group that does not'. | We will require two minimum criteria for the inclusion of CBA studies. ► Data collection: we will include CBA studies if the data for the intervention and control groups were collected prospectively in the same time frame. ► Choice of control: we will include CBA studies that include a control at a second site to avoid contamination of the intervention to the control group if the settings and populations are the same for the intervention and control groups. |

The definitions of study designs were adopted from The Cochrane Effective Practice and Organization of Care group.[33]
CBA, controlled before and after; NRCT, non-randomised controlled trial; RCT, randomised controlled trial.

## Comparison

For the primary objective of the review, the comparison group will be children receiving RUTF containing iron at the current WHO-recommended level of 1.9 mg/100 kcal (10–14 mg/100 g) to treat SAM. For the secondary objective of the review, we will consider studies where different types/doses/techniques of RUTF are studied to improve the bioavailability of iron even though the iron content may be the same in all kinds of RUTF given to children. We will exclude studies where the comparison group did not receive any RUTF.

## OUTCOMES
### Primary outcomes

1. Blood haemoglobin concentration (continuous outcome).
2. Any anaemia (dichotomous outcome, as defined by the authors).
3. Severe anaemia (dichotomous outcome, as defined by the authors).
4. Iron deficiency anaemia (dichotomous outcome as defined by the authors).
5. Recovery from SAM (dichotomous outcome, as defined by the authors).
6. Any adverse events (dichotomous outcome).

### Secondary outcomes

1. Serum ferritin level (continuous outcome).
2. Serum zinc level (continuous outcome).
3. Serum copper level (continuous outcome).
4. Serum iron level (continuous outcome).
5. Adverse events: malaria: (as defined by the authors) (dichotomous outcome).
6. Relapse (dichotomous outcome).

7. All-cause mortality (dichotomous outcome).
8. Admission to an inpatient facility. (dichotomous outcome).
9. Withdrawal from the trial (dichotomous outcome).
10. Constipation (<3 bowel movements per week) (dichotomous outcome).
11. Adverse events: diarrhoea (>3 loose stools per day) (dichotomous outcome).
12. Adverse events: pneumonia (as defined by authors) (dichotomous outcome).
13. Weight for age (kg or Z scores).
14. Height for age (cm or Z scores)
15. Weight for height Z score
16. Microbiome outcomes: alpha diversity and beta diversity
17. Neurodevelopmental outcomes (continuous outcome) at 1 year and the longest follow-up.

The term neurodevelopment is a composite term that refers to cognitive, neurological and/or sensory outcomes. This assessment may include intellectual disability as measured on the Mental Developmental Index of the Bayley Scales of Infant Development, gross motor delay measured on the Gross Motor Function Classification System, and so on.

We will consider the data for all the above outcomes at the longest follow-up.

## LITERATURE SEARCH

We will conduct systematic electronic queries using key terms in several databases, including PubMed, EMBASE, the Cochrane Central Register for Controlled Trials, Web of Science, CINHAL,

Scopus, LILACS and WHO Global Index Medicus. No search restrictions will be used to exclude studies based on the outcome, publication year, publication status or language. The references of formerly published reviews and recently published studies will be examined for potential inclusion. We will also use the citation tracking function of the included studies in the PubMed to look for any eligible studies. In addition to the above resources, ClinicalTrials.gov will be used to identify studies currently underway. We will also use the clinical trial registration number to find all the relevant studies published from a particular trial. Finally, we will search the grey literature and search the websites of pertinent international agencies such as WHO (including WHO's Reproductive Health Library, electronic Library of Evidence for Nutrition Actions and Global database on the Implementation of Nutrition Action), UNICEF, Global Alliance for Improved Nutrition, International Food Policy Research Institute, International Initiative for Impact Evaluation (3ie), Nutrition International, World Bank, USAID and affiliates (eg, FANTA, SPRING) and the World Food Program. We will also search the abstracts presented in major paediatric conferences such as Pediatric Academic Society meeting. Proposed search strategies for different electronic databases are shown in online supplemental appendix 1.

## DATA EXTRACTION AND SYNTHESIS
### Selection of studies
Searches from all the databases will be combined in bibliographic software (EndNote), and duplicates will be removed. Two authors will use a three-phase approach in duplicate to screen studies identified from the search for eligibility. In the first phase, authors will screen the titles and abstracts to identify potentially eligible ones; studies selected during this initial phase will then go on to a full-text review as the second phase. Lastly, studies determined to be eligible subsequent to full-text review will undergo data extraction during the third and final phase. The software Covidence V.14[26] will be used to assist the screening process. Two authors will independently extract data from screened studies and compare their findings. Potential conflicts will be resolved through discussion, and the senior author on the team may assist as needed to resolve any conflicts. If a study is only available in abstract form, we will write to authors to obtain details on methods and results. If we cannot get full details of the study methods available in the abstract, we will decide about inclusion based on details available in the abstract. If a study is available in a language other than English, we will attempt to complete the translation using local resources. If a study was published in more than one report (multiple publications), we will count those multiple reports as a single study and extract information from all the available reports as needed.

### Data extraction
A data extraction form was designed to support the review and data extraction process (online supplemental document 1). Two authors will independently extract the data and compare their findings. Any conflict will be resolved by discussion and with the help of the senior author on the team if needed. The following information will be extracted for each study where available: study design (randomised controlled trial, quasirandomised experimental design or cohort study, CBA study), study site (country/region), study year, study type, intervention (dose of iron and dose of RUTF, duration, frequency and composition), exposures, comparison, outcomes, whether the results were adjusted for confounders and risk of bias. To avoid reviewer bias, we are deciding a priori the order of preference for extracting outcomes when data might be available in several formats. For randomised studies, we will prefer data that require the least manipulation by authors or inference by reviewers. We will extract the raw values (eg, means and SDs) rather than calculated effect sizes (eg, Cohen's d). For mortality data, we will give preference to denominators in the following order: number with the definite outcome known (or imputed as described below), number randomised and child years. For morbidity outcomes to which both survivors and non-survivors may have contributed data (eg, the incidence of pneumonia), we will give preference to child years, number with the definite outcome known, and number randomised.

### Studies with missing data
We will document attrition during data extraction. If data are missing for some cases, or if reasons for dropout are not reported, we will contact the trial authors to request the full data. If the authors report the missing data and report results using imputations for the missing data, we will use the latter. If a study does not report the SD for the continuous outcome and the SD cannot be calculated from the reported data (such as SE, CI, p value), we will write to the authors to request the data for SD. If the SD data are not available from authors, we will use SD from a similar study that has a similar study population. We will prefer to use the final values of a continuous outcome for a given follow-up. If the final values are not available but the difference between the end and the start of the study, we will write to authors to request the final values. If the final values are not available, we will use the difference or rate of change. We will use the data based on intention-to-treat analysis. If the data for intention-to-treat analysis is not given in the study, we will create our own intention-to-treat analysis for participants with known outcomes. If there is significant attrition between the randomised participants versus participants completing the study, we will include such a study but will investigate further with sensitivity analysis.

### Assessment of risk of bias in included studies
The risk of bias in studies will be evaluated using the Cochrane risk of bias (ROB 2.0) and Cochrane ROBINS-I for non-randomised and observational studies.[27 28] Two review authors will independently assess and agree on

**Table 2** Sample table to describe the studies that address delivery mechanism to increase the bioavailability of iron via RUTF in children 6 months and older with severe acute malnutrition

| Study | Study participants | Formulations used | Comparison group | Outcomes | Notes |
|---|---|---|---|---|---|
| - | | | | | |
| - | | | | | |
| - | | | | | |

RUTF, ready-to-use therapeutic food.

the risk of bias for the individual studies for an outcome. The discussion will resolve any disagreements, and if no agreement can be made, the senior review author will be consulted. We will assess the effect of assignment to intervention (the intention-to-treat effect) for randomised trials by addressing five domains of signalling questions in ROB-2 including: bias arising from the randomisation process, bias due to deviations from intended interventions, bias due to missing outcome data, bias in the measurement of the outcome and bias in the selection of the reported result. Each domain will receive a ranking of one of the following: low risk of bias, some concerns of bias, or high risk of bias. We will include quotes from the study for each signalling question as evidence for our ranking decision. The overall risk of bias will be determined based on the lowest ranking for individual domains. For example, if only one domain, 'some concerns,' is ranked, then the overall risk of bias will be 'some concerns'.

### Data synthesis

The findings from the systematic review will be reported both qualitatively and quantitatively. A narrative synthesis will be used to report the characteristics of all included studies. We will also narratively describe the data in a table for the secondary objective of this review (table 2). A random-effects meta-analysis will be employed when at least two studies possess sufficient clinical and methodological uniformity for synthesis for the primary objective of the review. We will use the generic inverse method to pool the studies in the meta-analysis. The software RevMan will be used for statistical analysis.[29] Dichotomous outcomes will be assessed using relative risk effect sizes and presented with 95% CIs. In the case of morbidity outcomes, we will combine all available data whenever possible if outcomes are measured in different ways. For example, we will include all types of diarrhoea (mild, moderate and severe) as a dichotomous value (yes/no) if participants had greater than 3 instances of loose stools per day. We will include the occurrence of anaemia, iron-deficiency anaemia and pneumonia throughout the study as dichotomous values (yes/no). We will pool the data for continuous outcomes to obtain an average mean difference and report it with its 95% CI. If data are reported in different units (eg, few studies report weight in kg and the others report in Z scores), we will use a standardised mean difference effect size and report it with its 95% CI. We will pool the data separately from randomised and

observational studies. If an observational study reports both adjusted and unadjusted values, we will use the adjusted values for meta-analysis.

We will consider the following pairs of comparisons:

High iron content in RUTF versus WHO standard iron content in RUTF

The Grading of Recommendations Assessment, Development and Evaluation (GRADE) approach will be used to evaluate overall evidence quality using the software GRADEpro.[30] The GRADE approach is a comprehensive framework used to assess the overall quality of evidence for an outcome using characteristics such as study design, heterogeneity, directness of evidence, risk of bias, publication bias and precision of effect estimates.[31] The results of the GRADE assessment will be included in a summary of the findings table. The table will contain quality ratings characterised as very low (we have very little confidence in the effect estimate), low (we have limited confidence in the effect estimate), moderate (we have moderate confidence in the effect estimate; the true effect is likely close to the estimate of the effect) or high (we have high confidence that the true effect lies close to that of the estimate of the effect) based on the primary outcomes of each study (table 3). The study will be started on 18 February 2022, and hope to be completed by 18 April 2022.

### Patient and public involvement

No patient or public involvement was considered in the preparation of this protocol.

## UNIT OF ANALYSIS ISSUES

### Multiple-arm trial

Studies with multiple treatment arms will be included if eligible. For multiple-arm trials, we will include data so that the only difference between the groups is the use of iron in RUTF.

### Cluster trials

Cluster assignment trials will be analysed together with individually randomised trials. We will use the cluster-adjusted values; if the trial results are not adjusted for cluster design, we will adjust the result by methods given in the Cochrane[32] handbook for systematic reviews.

### Assessment of heterogeneity

Statistical heterogeneity of effect sizes within any given meta-analysis will be assessed using the $\chi^2$, $I^2$ and tau statistics. We will assess statistical heterogeneity by visual

**Table 3** The criteria for the Grading of Recommendations Assessment, Development, and Evaluation approach to examine the overall certainty of evidence[31]

| Study design | Quality of evidence | Lower certainty score if | Higher certainty score if |
|---|---|---|---|
| Randomised trial | High | Risk of bias<br>► 1 Serious<br>► 2 Very serious<br>Inconsistency<br>► 1 Serious<br>► 2 Very serious | Large effect<br>+1 Large<br>+2 Very large<br><br>Dose–response<br>+1 evidence of a gradient |
| | Moderate | | |
| Observational study | Low | Indirectness<br>► 1 Serious<br>► 2 Very serious<br>Imprecision<br>► 1 Serious<br>► 2 Very serious<br>Publication bias<br>► 1 Likely<br>► 2 Very likely | All plausible confounding would:<br>+1 Reduce a demonstrated effect<br><br>+1 Suggest a spurious effect when results show no effect |
| | Very low | | |

inspection of forest plots, performing the $\chi^2$ test (assessing the p value), and calculating the $Tau^2$ and $I^2$ statistics. Statistical heterogeneity will be considered significant if the p value is <0.10, $I^2$ value exceeds 50%, and the examination of forest plots shows substantial variability in the effect of the intervention. We will perform subgroup analysis to determine the reasons for any identified statistical heterogeneity.

## Assessment of reporting bias

Small study and publication bias will be assessed using funnel plots and regression tests for funnel plot asymmetry when a meta-analysis includes at least 10 studies.

## Subgroup analyses

1. Settings: country—low-income country versus middle-income country versus high-income country.
2. Type of RUTF: standard RUTF versus non-standard.
3. Role of RUTF: RUTF as main treatment versus RUTF as a supplementary intervention.
4. Type of participants: studies that included children with HIV versus studies with children without HIV.
5. Age: <24 months versus 24–59 months versus >59 months.
6. Hospitalisation: children hospitalised (due to medical complication) prior to starting RUTF versus children not hospitalised prior to starting RUTF.
7. Iron compound (formulation/chemical compound and amount).
8. Dose: the intervention groups with a dose of iron greater than the standard WHO RUTF versus intervention group receiving dose lower than the standard WHO RUTF.
9. Anaemia status: children with anaemia at the baseline versus without anaemia at the baseline.
10. Time of follow-up: 1 month versus 3 months versus 6 months follow-up and the longest follow-up.

## Sensitivity analyses

1. Studies with a high risk of bias.
2. Random vs fixed-effect model.

## Amendments

We will do the literature searches, screening of titles, selection of studies, data extraction and analysis according to the aforementioned plan described in this protocol. If we do any additional analysis or change any of the a priori strategies, we will clearly describe that in the Methods and analysis section.

## Consent for publication

The authors give consent for the publication of the review.

## ETHICS AND DISSEMINATION

This is a systematic review and will not involve direct contact with human subjects. The findings of this review will be published in a peer-reviewed journal and will guide the WHO's recommendation on the optimal iron content in RUTF for the treatment of SAM in children aged 6–59 months.

**Author affiliations**
[1]Department of Pediatrics, Division of Pediatric Gastroenterology, Hepatology and Nutrition, SUNY Upstate Medical University, Syracuse, New York, USA
[2]College of Medicine, SUNY Upstate Medical University, Syracuse, New York, USA
[3]Library Sciences, SUNY Upstate Medical University, Syracuse, New York, USA
[4]College of Education, University of Oregon, Eugene, Oregon, USA
[5]Department of Pediatrics, Aga Khan University, Karachi, Pakistan
[6]Center of Excellence in Women and Child Health, The Aga Khan University, Karachi, Pakistan

**Acknowledgements** We are very thankful to Allison Daniel, Jaden Bendabenda, Lisa Rogers and Maria De Las Nieves Garcia Casal for their input to improve this protocol.

**Contributors** AI and MF wrote the first draft of the manuscript. AS and OT developed the search strategy. ET-S provided the support for methodology. ZAB, FFC and JKD provided the support for the content being evaluated in the review. All authors edited and reviewed the manuscript. AI is the guarantor of the review.

**Funding** This work is funded by World Health Organization grant no 202712614. WHO provided technical support for this work and guided the development of PICO (Population, Intervention, Comparison, Outcome) question for this review.

**Competing interests** None declared.

**Patient consent for publication** Not applicable.

**Provenance and peer review** Not commissioned; externally peer reviewed.

**ORCID iDs**
Aamer Imdad http://orcid.org/0000-0002-7026-0006
Emily Tanner-Smith http://orcid.org/0000-0002-5313-0664
Jai K Das http://orcid.org/0000-0002-2966-7162

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
