## [Reviewer comments · BMJ Open]

ARTICLE DETAILS

TITLE (PROVISIONAL)	Optimal iron content in ready-to-use therapeutic foods for the treatment of severe acute malnutrition in the community settings. A protocol for the systematic review and meta-analysis
AUTHORS	Imdad, Aamer; François, Melissa; Chen, Fanny F; Smith, Abigail; Tsistinas, Olivia; Tanner-Smith, Emily; Das, Jai K.; Bhutta, Zulfiqar

VERSION 1 – REVIEW

REVIEWER	Kangas, Suvi T. International Rescue Committee, Airbel Impact Lab
REVIEW RETURNED	09-Oct-2021

GENERAL COMMENTS	Congratulations for the very clear protocol. Your review will address a key concern in related to the optimisation of malnutrition treatment. I only have minor comments for your consideration as follows: - In the article summary: would be great to mention the aim of the review and the studied outcomes- Studied outcomes: I would think looking at length of stay in treatment and weight gain velocity would be of interest for this study- Studied sub-groups: Could you look at outcomes by baseline iron status? Or baseline morbidity status? Or inflammation status? Malaria endemic versus not?- Timepoint: I wonder whether it would be sufficient to look at outcomes at discharge and maybe just use different duration of treatment as a sub-group?)- Dose: when talking about dose, be clear whether this is dose of RUTF or dose of iron in RUTF (in other words, iron fortification level). Since I am not familiar with meta-analysis statistics, I have requested a separate statistical review of the protocol. Good luck with the manuscript!
---

REVIEWER	Pastor, Rosario UCAVILA, Faculty of Health Sciences
REVIEW RETURNED	09-Jan-2022

GENERAL COMMENTS	In this protocol of systematic review and meta-analysis the authors will deal with an interesting topic once they carry out such a review and meta-analysis. However, I believe that publishing only the
--

	protocol is insufficient, and that publication should be made when such a review and meta-analysis has been conducted. Below, I make several recommendations that can be taken into account when deciding to tackle the entire work. ABSTRAC 1. In the abstract there are abbreviations that have not previously been indicated with the full name.  - The authors report that this systematic review was commissioned by WHO. Please clarify this issue with certificate reference where this assignment to the authors is recorded. BACKGROUND 1. I believe that this section should be improved/expanded by addressing aspects such as more extensive information on the RUTF formulations standards, etc.  - Furthermore, I believe that the need for this review should be better justified. What reviews are currently available on the subject? Why is this review necessary and what differentiates it from the others currently published? METHODS AND ANALYSIS  - The authors report that it will record the protocol of its review/meta-analysis in PROSPERO, but I think this should have already been done since the protocol is this that the authors intend to publish. - LITERATURE SEARCH. How will grey literature be treated? - What is the search strategy to use? - The authors report that a data extraction form will be designed. Creo que este formulario se debería mostrar en esta publicación (en anexos o material complementario). In addition, I believe that it should be expressly checked and detailed in this form if everything that the authors reflect in Primary outcomes and Secondary Outcomes is included in said form. To do this, it would be very helpful if the authors showed the form already designed or, failing that, the tables (in the absence of completion) that are going to be included in the review. I think more references should be provided in the article; 25 references I think are few to support and justify this protocol.
--	---

REVIEWER	Lubree, Himangi KEM Hospital Pune Research Centre, Vadu Rural Health Program
REVIEW RETURNED	11-Jan-2022

GENERAL COMMENTS	There is a need for a systematic review for the concerned topic and I appreciate the authors for taking up the task to do so. 1) The authors do have clarity on the outcomes but the methods and analysis section. 2) The population section is very confusing and needs clarity. Also as the objective mentions uncomplicated acute severe malnutrition cases, studies with comorbidities eg: HIV and malaria should be excluded as these are disease conditions leading to undernutrition 3)The intervention section if mentioned pointwise or tabular would present more clarity 4) If the study selection process in explained in a stepwise manner it would be helpful to understand
---

VERSION 1 – AUTHOR RESPONSE

Reviewer: 1

Dr. Suvi T. Kangas, International Rescue Committee Comments to the Author:

Congratulations for the very clear protocol. Your review will address a key concern in related to the optimisation of malnutrition treatment.

Response: We thank you for your time and useful feedback.

I only have minor comments for your consideration as follows:

- In the article summary: would be great to mention the aim of the review and the studied outcomes

Response: Thank you for this comment. We added the aims of the review and studies outcomes in the summary

- Studied outcomes: I would think looking at length of stay in treatment and weight gain velocity would be of interest for this study

Response: Thank you for this comment: These are very important observation. We measure the weight gain by the following two outcomes: Recovery (which is defined based on changes in weight for height) and weight for age that indirectly measures the weight gain velocity. We incorporate the length of stay with the aim to measure the outcome at 1, 6 and 12 months and the longest follow up. We include the outcome of 'withdrawal from trial' which indirectly measure any unwanted effect of intervention leading to increased withdrawal in one study groups vs. other. We also include relapse that indicates the effect of treatment based on time.

- Studied sub-groups: Could you look at outcomes by baseline iron status? Or baseline morbidity status? Or inflammation status? Malaria endemic versus not?

Response: Thank you for this comment. We agree and have planned the following subgroup analyses to cover the aspects mentioned above.

“- Settings: Country: Low-income country vs. middle-income country vs. high-income country

- Type of RUTF: Standard RUTF vs. Non-standard

- Role of RUTF: RUTF as main treatment vs. RUTF as a supplementary intervention

- Type of participants: Studies that included children with HIV vs. Studies with children without HIV.

- Age: <24 months vs. 24-59 months vs. > 59 months

- Hospitalization: Children hospitalized (due to medical complication) prior to starting RUTF vs. Children not hospitalized prior to starting RUTF

- Iron compound (formulation/chemical compound and amount)

- Dose: The intervention groups with a dose of iron > standard WHO RUTF Vs. Intervention group receiving dose lower the Standard WHO RUTF

-Children with anemia at the baseline vs. Children without anemia

- Time of follow up: 1 month vs. 3 months vs. 6 months follow up and the longest follow up”

- Timepoint: I wonder whether it would be sufficient to look at outcomes at discharge and maybe just use different duration of treatment as a sub-group?)

Response: Thank you. This is a very important observation and we agree with this suggestion. We changed the outcomes to longest follow up (which will be at discharge) and consider other follow up times in a subgroup analysis as mentioned above (the last subgroup analysis).

- Dose: when talking about dose, be clear whether this is dose of RUTF or dose of iron in RUTF (in other words, iron fortification level).

Response: Thank you. This is a very important observation and we clarified between the dose of RUTF and the dose of iron in RUTF, where applicable.

Since I am not familiar with meta-analysis statistics, I have requested a separate statistical review of the protocol.

Response: Thank you for your comment. We welcome any feedback on statistics of the meta-analysis.

Good luck with the manuscript!

Response: Thank you

Reviewer: 2

Dr. Rosario Pastor, UCAVILA

Comments to the Author:

In this protocol of systematic review and meta-analysis the authors will deal with an interesting topic once they carry out such a review and meta-analysis. However, I believe that publishing only the protocol is insufficient, and that publication should be made when such a review and meta-analysis has been conducted.

Below, I make several recommendations that can be taken into account when deciding to tackle the entire work.

Response: Thank you for your kind feedback. We appreciate your time to review our protocol. As this work will feed into guideline development at WHO, we wanted to make the process of evidence synthesis very transparent and make our plan of analysis and methods of review publicly available before the study was conducted. We therefore think that publication of the protocol is justified. We highly appreciate your feedback and have modified the manuscript based on the comments.

ABSTRACT

1. In the abstract there are abbreviations that have not previously been indicated with the full name.

Response: Thank you for pointing this out. The Abbreviations SAM and WHO were not defined and we have revised the text to include the text for these abbreviations.

- The authors report that this systematic review was commissioned by WHO. Please clarify this issue with certificate reference where this assignment to the authors is recorded.

Response: Thank you for asking this question. We provide the grant number in the funding section of the protocol as follows, "This work is funded by World Health Organization grant no 202712614. WHO also provided technical support for this work."

BACKGROUND

1. I believe that this section should be improved/expanded by addressing aspects such as more extensive information on the RUTF formulations standards, etc.

Response: Thank you. We agree and we have revised the introduction section and expanded the text.

- Furthermore, I believe that the need for this review should be better justified. What reviews are currently available on the subject? Why is this review necessary and what differentiates it from the others currently published?

Response: Thank you. We agree and we have added text to justify the conduct of this study.

METHODS AND ANALYSIS

- The authors report that it will record the protocol of its review/meta-analysis in PROSPERO, but I think this should have already been done since the protocol is this that the authors intend to publish.

Response: Thank you for asking about PROSPERO registration. Since the submission of this protocol to BMJ open, we have completed the registration at PROSPERO and include the ID (ID=CRD42021278006) in the manuscript now

- LITERATURE SEARCH. How will grey literature be treated?

Response: Thank you for asking this question. This was not very clearly stated in the protocol. We added the clarification and layout the plan for search for the grey literature as follows, "Finally, we will search the grey literature and search the websites of pertinent international agencies such as WHO (including WHO's Reproductive Health Library, electronic Library of Evidence for Nutrition Actions and Global database on the Implementation of Nutrition Action), UNICEF, Global Alliance for Improved Nutrition, International Food Policy Research Institute, International Initiative for Impact Evaluation (3ie), Nutrition International, World Bank, USAID and affiliates (e.g., FANTA, SPRING), and the World Food Program. We will also search the abstract presented in major pediatric conference such as Pediatric academic society meeting."

- What is the search strategy to use?

Response: Thank you for asking about the search strategy. We had provided the search strategy in a supplementary document which we now include in the main protocol as appendix 1.

- The authors report that a data extraction form will be designed. I think this form should be shown in this publication (in annexes or supplementary material). In addition, I believe that it should be expressly checked and detailed in this form if everything that the authors reflect in Primary outcomes and Secondary Outcomes is included in said form. To do this, it would be very helpful if the authors showed the form already designed or, failing that, the tables (in the absence of completion) that are going to be included in the review.

Response: Thank you for asking about the data extraction sheet. We agree and now provide the Excel data extraction sheet in the form of supplementary document.

I think more references should be provided in the article; 25 references I think are few to support and justify this protocol.

Response: We agree. We have added additional references and the count now is 33

Reviewer: 3

Dr. Himangi Lubree, KEM Hospital Pune Research Centre Comments to the Author:

There is a need for a systematic review for the concerned topic and I appreciate the authors for taking up the task to do so.

Response: Thank you for your time and we appreciate your efforts to review the protocol.

1) The authors do have clarity on the outcomes but the methods and analysis section.

Response: Thank you. We have revised these sections as advised.

2) The population section is very confusing and needs clarity. Also as the objective mentions uncomplicated acute severe malnutrition cases, studies with comorbidities eg: HIV and malaria should be excluded as these are disease conditions leading to undernutrition

Response: Thank you. We agree and have revised this section as follows.

"The population of interest is children age 6 months or older with severe acute malnutrition being managed in the outpatient settings. The definition of severe acute malnutrition will be based on weight for height z scores (< -3 SD for WHO growth standards) or mid-upper arm circumference (< 115 mm)

or presence of bilateral edema¹⁷. We will consider studies that included children who were previously admitted to hospital and are now being rehabilitated in the community settings with the help of RUTF. We will consider studies that included children infected with human immunodeficiency virus (HIV) or Malaria. We will exclude studies of children with complicated SAM who are admitted to the hospital, as those participants might have complications such as pneumonia or severe diarrhea. We will exclude studies done specifically on participants who have chronic diseases, genetic disorders, or congenital anomalies.”

We however decided to keep the children infected with HIV and Malaria. This is because the prevalence of malnutrition is high in patients with HIV. We however do a subgroup analysis to look at the population with and without HIV. We also kept the children with Malaria as there is a concerns that the risk of morality will be increased with higher doses of iron so we did not want to miss that aspect of the study question.

3)The intervention section if mentioned pointwise or tabular would present more clarity

Response: Thank you for pointing this out. We agree that this section can be improved. We have revised the text as follows:

“The intervention of interest is the dose of iron in RUTF other than in the standard RUTF for the treatment of severe acute malnutrition in community settings. We will include studies of iron-fortified RUTF and exclude those studies where iron was given separately as supplements, such as tablets, syrup, multiple micronutrient powder, etc. We will include studies where RUTF is given as the main intervention, and it met total daily requirements or was given as a supplementary intervention to the usual diet for children with severe acute malnutrition recovering at home. WHO recommends that a standard RUTF should have at least 1.9 mg/100 kcal (or 10-14 mg/100 g) of iron to treat severe acute malnutrition in children 6-59 months of age¹⁷. We will include studies irrespective of the type of RUTF used, i.e., standard RUTF vs. low milk-based vs. non-milk-based vs. locally prepared. We will exclude studies where RUTF was given to healthy children for the prevention of severe acute malnutrition; treatment of moderate acute malnutrition; as part of complementary feeding.”

4) If the study selection process in explained in a stepwise manner it would be helpful to understand
Response: Thank you for this comment. We include the step wise approach in the manuscript as follows

“Two authors will use a three-phase approach in duplicate to screen studies identified from the search for eligibility. In the first phase, authors will screen the titles and abstracts to identify potentially eligible ones; studies selected during this initial phase will then go on to a full-text review as the second phase. Lastly, studies determined to be eligible subsequent to full-text review will undergo data extraction during the third and final phase.”

VERSION 2 – REVIEW

REVIEWER	Lubree, Himangi KEM Hospital Pune Research Centre, Vadu Rural Health Program
REVIEW RETURNED	12-Feb-2022

GENERAL COMMENTS	The authors have addressed the reviewer's comments appropriately. One comment is regarding the timelines of the project. Page 14 line 28 says January 2022 to March 2022 which may not be applicable as we have already crossed the start timeline
---

VERSION 2 – AUTHOR RESPONSE

Reviewer: 3

Dr. Himangi Lubree, KEM Hospital Pune Research Centre Comments to the Author:

The authors have addressed the reviewer's comments appropriately. One comment is regarding the timelines of the project.

Page 14 line 28 says January 2022 to March 2022 which may not be applicable as we have already crossed the start timeline

Response: Thank you for this comment. We updated the date of start of the project

Reviewer: 3

Competing interests of Reviewer: non